# Iron Chelator DIBI Suppresses Formation of Ciprofloxacin-Induced Antibiotic Resistance in *Staphylococcus aureus*

**DOI:** 10.3390/antibiotics11111642

**Published:** 2022-11-17

**Authors:** David S. Allan, Bruce E. Holbein

**Affiliations:** Fe Pharmaceuticals Canada Inc. #58, The Labs at Innovacorp, 1344 Summer Street, Halifax, NS B3H O8A, Canada

**Keywords:** ciprofloxacin, SCV, resistance, Iron, DIBI, ROS

## Abstract

Antibiotic resistance of bacterial pathogens results from their exposure to antibiotics and this has become a serious growing problem that limits effective use of antibiotics. Resistance can arise from mutations induced by antibiotic-mediated damage with these mutants possessing reduced target sensitivity. We have studied ciprofloxacin (CIP)-mediated killing of *Staphylococcus aureus* and the influence of the Reactive Oxygen Species (ROS) inactivator, thiourea and the iron chelator DIBI, on initial killing by CIP and their effects on survival and outgrowth upon prolonged exposure to CIP. CIP at 2× MIC caused a rapid initial killing which was not influenced by initial bacterial iron status and which was followed by robust recovery growth over 96 h exposure. Thiourea and DIBI did slow the initial rate of CIP killing but the overall extent of kill by 24 h exposure was like CIP alone. Thiourea permitted recovery growth whereas this was strongly suppressed by DIBI. Small Colony Variant (SCV) survivors were progressively enriched in the survivor population during CIP exposure, and these were found to have stable slow-growth phenotype and acquired resistance to CIP and moxifloxacin but not to other non-related antibiotics. DIBI totally suppressed SCV formation with all survivors remaining sensitive to CIP and to DIBI. DIBI exposure did not promote resistance to DIBI. Our evidence indicates a high potential for DIBI as an adjunct to CIP and other antibiotics to both improve antibiotic efficacy and to thwart antibiotic resistance development.

## 1. Introduction

Exposure of bacteria to antibiotics is known to trigger emergence of antibiotic resistant persister cells which may represent a minor sub-population of an initially present, resistant phenotype that becomes positively enriched by antibiotic selection [1,2]. Alternatively, antibiotic resistant Small Colony Variants (SCVs) can arise from accelerated mutations triggered by DNA damage resulting from exposure to an antibiotic [3]. SCV isolates obtained from clinical specimens following treatment of infected human patients with β-lactams and aminoglycosides often have menadione or hemin auxotrophy, whereas thymidine auxotrophy is typically seen in SCVs from patients treated with anti-folates [4]. Importantly, regardless of the initial drug treatment which induced the SCV phenotype, these variants can often have cross-resistance to other unrelated antibiotics due to their altered metabolism and electron gradient deficiency [5,6,7].

The SCV response to antibiotics has been regarded as part of the generalized SOS response as observed from stress caused by various agents including antibiotics [8,9]. Interestingly, intracellular free iron plays a role in the generation of Reactive Oxygen Species (ROS) but bacterial defense against ROS also requires catalase and superoxide dismutase both of which are iron-dependent enzymes. Additionally, repair of DNA damage involves ribonucleotide reductase which is another enzyme that is iron dependent for activity. Thus, iron requirements and bacterial iron regulation can affect antibiotic resistance development and expression at several discrete physiological levels and these aspects have been reviewed recently [10].

Bactericidal antibiotics such as CIP can cause oxidative stress that can damage iron-sulfur clusters [11,12]. Substantial evidence that CIP induces mutations leading to resistance via ROS- and iron-dependent mechanisms has been obtained in *Escherichia coli*, where Δ*fur* and Δ*sodAB* knockout mutants, with loss of iron homeostatic control leading to accumulation of intracellular iron, or loss of superoxide dismutase activity, respectively, both had significantly higher rates of CIP resistant mutations following exposure compared to wild-type [13]. CIP and related structures are also known to bind iron [14,15], and thus could bring iron into the bacterial cytoplasm or mobilize iron within it. This aspect has not been investigated.

CIP has been shown to be mutagenic in *Staphylococcus aureus* (*S. aureus*), increasing both mutation rates and DNA recombination, via activation of RecA, which functions directly in DNA repair, as well as autocleavage of repressor molecule LexA, allowing the induction of the SOS response, including the error-prone UmuC-type polymerase [16,17]. Topoisomerase IV (GrlA) is the primary target of CIP in Gram positive bacteria, and functions in assisting DNA replication (along with DNA gyrase, the secondary CIP target), thus interference with GrlA leads to DNA damage that requires repair [15]. RecA repairs DNA by forming a stabilized nucleoprotein filament, which then enables repair, a process which is made possible in Gram positive bacteria by AddAB, a nuclease which acts via an essential iron-sulfur cluster [18].

The above mechanisms leading to antibiotic resistance would all require DNA synthesis/repair and growth as well as metabolic and macromolecular synthesis activity and thus, would require sufficient supply of key iron-dependent enzymes. Therefore, iron sequestration sufficient to limit the adequate supply to these key iron-dependent enzymes might be expected to suppress bacterial repair and recovery from antibiotic-induced injury, and also prevent the expression of resistance mechanisms. Exploitation of this iron-dependence of antibiotic resistance with new iron-sequestering therapeutic agents provides a potential new avenue to address microbial antibiotic resistance.

Apo-transferrin, which is an iron- sequestering agent elaborated by the host during infection, has been reported to suppress formation of CIP-resistant persister clones in *Klebsiella pneumoniae* [19]. We have previously shown that the synthetic iron chelator DIBI [20] potentiates CIP activity for *S. aureus* through synergistic killing [21]. In this study we have further examined the nature of DIBI potentiation of CIP activity and its effects on the formation of CIP resistant survivors.

## 2. Results

### 2.1. Staphylococcus aureus Accumulates Excess Medium Iron

While MHB is considered a standard cultivation medium for MIC testing of antibacterial agents, it is also known to contain iron in large excess over the minimal microbial requirements of this growth-essential metal [21]. This has been shown to affect sensitivity to various iron chelators especially in the case of *S. aureus*, likely because this bacterium can store excess iron as supplied in the growth medium [22,23]. 

We assessed the cellular iron status of *S. aureus* as grown in MHB, MHB substantially stripped of its excess iron using FEC1 (FECMHB), and in RPMI, a fully defined chemical medium containing a low content of only inorganic iron sources. Cellular concentration of manganese was also determined as it has been shown to have similar electrochemical properties to iron and is also involved in bacterial ROS regulation [24]. Zinc was also determined for comparison as it is not known to be involved in bacterial ROS regulation. Growth in MHB promoted cellular storage of both excess iron and manganese but in contrast, not for zinc, as shown in Table 1. Cellular iron and manganese concentrations were both approximately 10X higher with cultivation in MHB versus RPMI, noting that RPMI has been shown to provide sufficient iron for unrestricted growth of *S. aureus* [21]. Growth in FECMHB resulted in substantially reduced cellular loading of both iron and manganese.

### 2.2. Cellular Iron Status Affects Sensitivity to DIBI but Not to CIP

We then tested sensitivity to both DIBI and CIP by measuring their MICs for *S. aureus* ATCC 25923 and ATCC 43300 as cultivated in these three different media with results as shown in Table 2.

Cellular iron status greatly affected sensitivity to DIBI. MHB-grown bacteria had a 2000–4000-fold increase in MIC for DIBI compared to FECMHB- or RPMI-grown bacteria. However, iron status did not significantly affect sensitivity to CIP, with CIP sensitivity remaining similarly high (MIC = 0.25–5.0 µg/mL) whether bacteria were loaded with excess iron or not.

We had previously shown that a low, i.e., near-MIC concentration of DIBI (2.5 µg/mL (0.28 µM)) together with 1XMIC CIP provided synergistic killing with a >4 log_10_ CFU reduction of FECMHB-grown *S. aureus* 43300 by 24 h exposure, compared to the CIP-alone treated cells which grew substantially over 24 h in the presence of this relatively low CIP concentration [21]. For the present studies, we increased CIP exposure to 2XMIC CIP and this resulted in rapid initial killing with an overall similar extent of killing by 24 h for both MHB- and FECMHB-cultivated ATCC 25923 as shown in Table 3. 

There was a >4 log_10_ killing by CIP by 24 h for both MHB- and FECMHB-grown cells indicating that bacterial cell loading of iron did not significantly affect 24 h overall killing by CIP, a result consistent with the MIC results of Table 1. Consequently, we decided to use MHB-grown bacteria in our model system of CIP exposure to ensure presence of excess cellular iron, given the possible role of excess iron in ROS-mediated CIP killing. MHB-grown inocula were transferred to RPMI to limit additional external iron supplies for CIP-exposure/recovery testing. While MHB- and FECMHB-grown bacteria had similar sensitivities to CIP, these had dramatically different sensitivities to DIBI as shown in Table 2, as compared to RPMI-grown cells and due to elevated endogenous iron stores. The DIBI MIC for MHB-grown ATCC 25923 was 2000 µg/mL (222 µM), i.e., 1000× higher than for FECMHB- or RPMI-grown cells. Therefore, we used a higher but still substantially sub-MIC concentration of DIBI (50 µg/mL (5.6 µM) for these experiments to partially restrict iron supply during CIP-exposure. At this concentration, DIBI on its own resulted in a somewhat reduced Y_max_ by 24 h and 48 h in comparison to untreated controls as shown in Figure 1. 

### 2.3. DIBI Suppresses Recovery Growth of CIP-Exposed Bacteria

In our test model, bacteria were exposed to CIP at 2XMIC to provide substantial initial killing of *S. aureus* ATCC 25923 and the kill cultures were then incubated in the continued presence of CIP over 48 h for periodic sampling of survivor growth. This antibiotic exposure was found to cause approximately a 4 Log_10_ CFU/mL initial killing and this initial killing phase by CIP was essentially complete by 12 h exposure with no significant change in survivor numbers until after 24 h (Figure 1). The surviving population, although still exposed to CIP, exhibited substantial recovery growth after 24 h with increasing survivor numbers through 48 h exposure.

Both DIBI and thiourea slowed the initial killing rate by CIP but the overall extent of killing with both DIBI and thiourea treated CIP-exposed cultures was similar to that for CIP alone by 24 h. In the case of DIBI-treated, CIP-exposed cultures, recovery growth after 24 h was impaired with the 48 h bacterial population still below the initial inoculum CFU/mL. Thiourea appeared to significantly enhance recovery growth of CIP-exposed cultures after 24 h.

We then repeated these sets of experiments but followed survivor growth for an extended 96 h exposure period with results as shown in Figure 2. Overall killing at 24 h by CIP was again similar for CIP alone and DIBI and thiourea treated CIP-exposed bacteria, despite the early reduced initial rate of killing for both thiourea and DIBI treated cultures. Thiourea promoted increased recovery after 24 h with survivor numbers increasing to those of untreated controls and for CIP alone by 96 h. DIBI treated, CIP-exposed survivor numbers remained low through the 96 h exposure with final CFU/mL remaining below the initial inoculum as shown in Figure 2. DIBI in combination with thiourea provided 24 h overall killing like that for DIBI alone but there was only modest recovery growth with DIBI-thiourea, i.e., compared to thiourea alone which allowed strong recovery growth similar to CIP alone by 96 h. DIBI-thiourea treated survivor numbers at 96 h CIP exposure remained below the initial inoculum numbers.

### 2.4. Prolonged CIP-Exposure Selective for Small Colony Variant Phenotype

All recovery enumeration plating for these CIP-exposure experiments was onto BA and we observed that in the case of CIP-exposed bacteria, a progressively increasing proportion of the recovered colonies over the recovery period had pin-point morphology and these required additional incubation to 48 h for reliable enumeration. These Small Colony Variants (SCVs) were not evident in any of the other treatment groups at 96 h.

Given recovery growth in the presence of 2× MIC CIP was substantially in the form of SCVs, we then compared the responses of two strains, ATCC 25923 and ATCC 43300 to this prolonged CIP exposure and the effects of DIBI on recovery growth and SCV formation. Both strains had similar overall responses to 2XMIC CIP and DIBI (Figure 3). The CIP-exposed bacteria recovered to near control numbers by 96 h. DIBI partially suppressed growth on its own and greatly suppressed recovery growth for both strains following CIP exposure. SCVs were enumerated for these experiments with results as shown in Figure 3. SCVs became progressively enriched in the CIP-exposed cultures for both strains between 48–96 h with approximately 80% of the total survivor population present as SCVs by 96 h exposure to CIP. No SCVs were present in cultures not exposed to CIP (Figure 3), including DIBI alone. Interestingly, when DIBI was combined with CIP it suppressed overall recovery growth and no survivors were recovered as SCVs over the 96 h exposure period.

The colony morphologies of the recovered SCVs for ATCC 25923 in comparison to non-SCVs from the survivor growth of the various treatments over the exposure period are shown in Figure 4. The progressive enrichment of SCVs in the CIP-exposed cultures over time can be seen, with colony morphology for all the other treatment groups exhibiting typical size variability for spot-plating. Similar results were obtained for ATCC 43300 (not shown).

### 2.5. CIP-induced persister SCVs Display Stable Acquired Resistance to Quinolones

Isolated sub-cultured clones, six each, as obtained from the various treatments at 96 h were confirmed to be *S. aureus* by growth on MSA and were then grown in RPMI. Their ability to grow in this simple medium, although more slowly, indicated these SCVs were neither auxotrophic nor otherwise fastidious. Growth was slow on MSA, BA, and the same basal medium (Trypticase Soy Agar) lacking blood, indicating that the slow growth was not related to the recovery medium. SCVs exhibited hemolysis on BA, although it was only apparent in heavily streaked areas or in cultures incubated for at least 48 h. Sub-cultured clones were grown and tested in RPMI for their sensitivities to DIBI, CIP, MOX, GEN, MUP and VAN with median 24 h MIC results as shown in Table 4.

CIP-induced SCV clones were all (12/12, 6 from each strain) found to have stable colony morphology after repeated subculture on antibiotic-free medium, suggestive of a genetic alteration as opposed to an epigenetic adaptation. These SCV clones all had an increased resistance to CIP with median MICs of 1.0 µg/mL (3 µM), with a range of 0.5–2.0 µg/mL, which is approximately 4–8× higher than those of untreated controls. Similar results of increased resistance were also found for MOX (ATCC 43300). However, representative survivor SCV clones remained as sensitive to MUP (ATCC 43300), GEN, and VAN (ATCC 25923) as controls (Table 4). DIBI-exposure alone did not result in any significant changes to CIP, MOX or GEN MICs in the recovered population. Interestingly, when DIBI was together with CIP, survivor clones were of normal colony morphology (Figure 4) and these had CIP and MOX MICs similar to controls (Table 4). DIBI sensitivities for recovered clones from all the treatments were similar with MICs of 4 µg/mL (0.4 µM). Non-SCV CIP survivors were also tested and had identical MIC results (not shown) to SCVs, notably with median CIP MIC of 1.0 µg/mL (3 µM) (range 0.5–1.0 µg/mL). As these were rare, it is unknown if they represent double mutants that have compensated for the slow growth phenotype, or a less common type of mutation that does not affect growth, or an epigenetic adaptation. Further genetic sequencing and phenotypic characterization would be required to determine the underlying mechanism of resistance in these CIP-treated SCV and non-SCV survivors.

## 3. Discussion

The killing action of second- and later-generation quinolone antibiotics such as CIP and MOX, respectively, are thought to involve at least two separate but inter-related pathways [25,26]. The primary pathway involves direct inhibition of the CIP targets, topoisomerase and DNA gyrase, which causes DNA strand breaks resulting in inhibition of bacterial replication and ultimately death. A secondary pathway involves a ROS-dependent mechanism of bacterial death induced by quinolone exposure. It is also thought that DNA strand break accumulation from the primary pathway ultimately leads to additional ROS production [25], to the point that ROS-mediated bactericidal activity has been suggested as the predominant cause of bacterial cell death during quinolone treatment [26,27]. However, the full extent of ROS involvement in antibiotic killing of bacteria remains somewhat controversial, as ROS detection with fluorescent probes may not be reliable given non-ROS antibiotic induced auto-fluorescence and, for example, the case of *Streptococcus pneumoniae* which is highly susceptible to killing by bactericidal antibiotics but yet lacks an electron transport chain, the presumed primary source of ROS [27].

We have used CIP at a relatively high concentration (2XMIC), and tested the effects of thiourea, a known ROS scavenger [10] during continued exposure to CIP. While thiourea slowed the initial rate of killing, the overall extent of killing by 24 h was like that for CIP alone. Others have reported that thiourea along with the iron chelator, dipyridyl, suppressed the ROS-dependent killing pathway in *Escherichia coli* treated with MOX [25]. While thiourea quenches existing ROS, dipyridyl would presumably lower ROS production by sequestering Fenton-ROS reactive iron. DIBI, which is a potent iron chelator and highly effective against *S. aureus* [20,21], was also found to slow the initial rate of CIP killing but not to lower the 24 h overall extent of kill during CIP-exposure. Thiourea in combination with DIBI did not further slow initial killing but DIBI prevented the post 24 h exposure recovery growth that we observed with thiourea alone.

Our results therefore suggest only a relatively small component of ROS-related initial killing by CIP in our model system. It is important to note in this regard that we tested CIP with fully replete/excess cellular iron stores, thereby ensuring excess intracellular iron for potential participation in ROS generation. Our previous studies of DIBI in combination with CIP had shown a >3 log_10_ synergy of *S. aureus* killing during exposure to a lower CIP concentration (1× MIC), thus suggesting that other iron-dependent systems independent of ROS production, are important for CIP activity [21]. Interestingly, intracellular iron-sulfur clusters are required for activity of bacterial DNA polymerase as needed for repair of antibiotic-induced DNA damage in *Staphylococcus* [28,29]. From this, it seems reasonable that DIBI might interfere with intracellular bacterial iron-sulfur cluster synthesis or other iron-containing targets as part of its mechanism of action and this possibility warrants further investigation.

We have found that CIP induces formation of SCV mutant survivors with stable acquired resistance to CIP and its related quinolone, MOX but not to the unrelated aminoglycoside, GEN or other antibiotics. This suggests that the slow growth phenotype seen here may be a consequence of CIP target-specific alteration, rather than a general metabolic change that affects bactericidal activity across antibiotic classes, as has been seen with other SCV phenotypes [7]. It is conceivable that if the topoisomerase or gyrase target(s) of CIP were spontaneously altered to reduce CIP activity, it could alter growth rates due to their role in DNA replication [15]. Further molecular characterization is required to determine the resistance mechanism.

Various antibiotics including quinolones are known to induce formation of bacterial persister survivor cells possessing resistance to antibiotics [1,30]. This response to CIP in *S. aureus* has been shown to occur, at least in part, through triggering of the SOS response via ROS generation and activation of the RecA system, and this can be inhibited by the herb baicalein [31], which has also been shown to be an iron chelator and inhibitor of the Fenton reaction [32].

Our findings that the ROS scavenger thiourea and the iron chelator DIBI both suppress SCV persister formation in response to CIP, indicating a role for ROS in causing genetic mutations that lead to antibiotic resistance. These presumptive SCV mutants were no more sensitive to DIBI than untreated clones, supporting the notion that DIBI prevented their initial formation, potentially through suppression of ROS-mediated DNA damage, which can increase DNA mutation, as DIBI alone did not suppress *S. aureus* growth over the full incubation period. This possibility is indirectly supported by a study using the cell-permeable iron chelator *o*-phenantroline with CIP-treated *E. coli* [13]. Though DIBI is expected to be impermeable to bacteria [20], it may similarly affect respiration (and thus ROS generation) through iron-sequestration.

The extracellular position of DIBI could be beneficial during treatment as it has been shown that the oft-studied turmeric extract curcumin, which enters mammalian cells and chelates iron [33,34], can have mixed results as an antibacterial agent (reviewed in [35]). For example, while curcumin inhibited *Listeria monocytogenes* and *Shigella flexneri*, it enhanced infections of mammalian cells by some other intracellular bacteria, including *Salmonella* species, *S. aureus*, and *Yersinia enterolytica* [36]. Furthermore, curcumin interfered with CIP treatment of *Salmonella* species by affecting host cell-signaling and respiratory burst [37]. Although *S. aureus* is generally extracellular in vivo, spontaneous SCV mutants can invade host cells, which provides a reservoir to evade host defences and antibiotic treatment (reviewed in [6]), thus preventing their formation during antibiotic treatment would be ideal.

Our findings that DIBI completely prevents the outgrowth of relatively quinolone-resistant *S. aureus* SCV survivors in multiple independent experiments with unrelated strains and restricts the overall recovery growth from CIP exposure are relevant in the context of bacterial evolution of antibiotic resistance and antibiotic failure, as it has been shown that resistance can be acquired in incremental steps [38]. The resulting MIC of CIP survivors was typically 1.0 µg/mL and as high as 2.0 µg/mL, which is at least double the concentration of 0.5 µg/mL encountered in the initial antibiotic challenge assay. It is important to note in this regard that the resistance increases we observed were from only a single CIP-exposure and these increased MICs, although ~4–8× higher than survivors of the control treatment, did not surpass the accepted MIC threshold (>1 µg/mL) for categorization as CIP-resistant using clinical breakpoints [39]. It would be interesting to re-expose these partially resistant clones to CIP to determine if repeated exposure would lead to categorical resistance, and what further effect iron-sequestration might play in suppressing development of resistance.

In the present study, we found no resistance development to DIBI upon prolonged exposure, in concordance with our previous findings of no resistance development upon repeated subculture of *S. aureus* in the presence of sub-MIC amounts of DIBI [40]. We have also previously shown that DIBI potentiates CIP efficacy for *Acinetobacter baumannii* during experimental infection with a CIP-resistant highly virulent clinical isolate [41]. Taken together, our evidence indicates a high potential for DIBI as an adjunct to CIP and other antibiotics to both improve antibiotic efficacy and to thwart antibiotic resistance development.

## 4. Materials and Methods

### 4.1. Antibiotics, Media, and Bacterial Strains

Ciprofloxacin (CIP), moxifloxacin (MOX), and mupirocin (MUP) (Sigma-Aldrich, St. Louis, MO, USA) were prepared as 1 mg/mL stocks in deionized water, and gentamicin (GEN) and vancomycin (VAN) (Sigma-Aldrich) were prepared at 10 mg/mL in water, and all antibiotics were stored at −80 °C, and thawed and diluted in RPMI just prior to use. The iron chelator DIBI-R12 (DIBI) (provided by Fe-Pharmaceuticals Canada Inc., formerly Chelation Partners Inc., Halifax, NS, Canada) was dissolved in Roswell Park Memorial Institute Medium 1640 (RPMI, Sigma-Aldrich) with l-glutamine, buffered with 0.165 M 3-(*N*-morpholino)-propanesulfonic acid (MOPS, Sigma-Aldrich), filter-sterilized and stored as 200 mg/mL or 20 mg/mL, at 4 °C. Thiourea (Sigma-Aldrich) was dissolved in RPMI at 100 mM, filter-sterilized and used fresh or from storage at 4 °C. DIBI was diluted in 10 mL RPMI in sterile glass flasks, to a final concentration of 50 µg/mL, CIP to 0.5 µg/mL (equivalent to 2× MIC as previously determined to be 0.25 µg/mL), and thiourea to 10 mM. An insoluble form of DIBI, called FEC-1 (Fe-Pharmaceuticals Canada Inc., Halifax, NS, Canada) was used to partially deferrate Mueller-Hinton Broth (MHB, Oxoid, Basingstoke, Hampshire, UK) prior to FEC-Fe removal by filtration, as previously described [42], to obtain FECMHB. The selective medium Mannitol Salt Agar (Sigma-Aldrich) was used for confirmation of *S. aureus*.

*Staphylococcus aureus* ATCC 43300, ATCC 6538 and ATCC 25923 clinical reference strains were obtained from Nova Scotia Health Authority (Halifax, NS, Canada) and grown from 10% glycerol preserved stocks at −80 °C on blood agar (BA, Oxoid, tryptic soy agar (TSA) containing 5% sheep’s blood), overnight at 35 °C.

### 4.2. Antibiotic Kill and Recovery Growth Testing

An isolated colony was used to inoculate 50 mL MHB liquid cultures in glass flasks, with shaking at 35 °C, 220 rpm, overnight. The optical density (600 nm) was measured, and used to create a standardized inoculum for addition to 10 mL RPMI flasks containing treatments or non-treated medium. The inoculum size was confirmed by performing serial dilutions and plate counting on BA.

At pre-determined time points, the optical density (OD) of each flask was measured at 600 nm by removing aliquots to be measured by the spectrophotometer. Colony counts were performed by doing serial dilutions in tubes containing phosphate-buffered saline (PBS, Sigma-Aldrich), followed by spot-plating 50 µL of appropriate dilutions onto antibiotic-free BA. Plates were incubated at 35 °C, and colonies were initially counted following 24 h incubation. Colonies which were noticeably smaller than normal (pin-point sized) were counted as a small colony variant (SCV), and their positions were marked. The count plates were returned to the incubator for an additional 24 h, and the plates were re-counted. Any new colonies not counted previously were counted as SCV, and tallied separately to be considered either together or apart from the total CFU/mL.

### 4.3. Isolation and Characterization of Treatment Survivors

Bacterial survivors from the CIP exposure experiments were selected from the 96 h time-point of the four treatment groups, and streaked onto fresh BA plates containing no antibiotic to assess the stability of the slow growth phenotype, and plates were incubated for 48 h to allow slow growers to reach a more normal colony size when necessary. Surviving colonies were also plated on MSA to rule out contamination, and collected growth was kept as frozen stocks in 10% glycerol.

### 4.4. Antibiotic Susceptibility

The minimum inhibitory concentration (MIC) was determined by the broth microdilution method as recommended by CLSI [43]. The re-cultured treatment survivors where inoculated into RPMI liquid tube cultures, and incubated 35 °C, 220 rpm, overnight. The OD of cultures was measured and used as a basis to make standardized suspensions of 0.1 OD/mL, which were then diluted further 1:10 in RPMI to produce an inoculum for MIC determinations. 96-well microplates containing serial 2-fold dilutions of 100 μL antibiotics were prepared containing antibiotic or DIBI, in duplicate wells, to which 5 μL of bacterial inoculum was added. Loaded microplates were incubated at 35 °C. MIC endpoints were read visually at 24 h and 48 h incubation as the minimum antibiotic concentration to allow zero visible growth within a well.

### 4.5. Trace Element Quantification of Iron, Manganese, and Zinc

The three strains of *S. aureus* were grown on BA for 24 h at 35 °C, prior to inoculation of an isolated colony into 50 mL MHB, 50 mL FEC-MHB, or 4 colonies into 100 mL RPMI in flasks. The cultures were incubated overnight at 35 °C with shaking at 220 rpm. Cultures were collected into 15 mL acid-washed polypropylene tubes and centrifuged at 3345 rcf, for 15 min, and the supernatant was decanted. Phosphate-buffered saline (PBS, pH 7.4, Sigma-Aldrich) was added to one tube and the pellets resuspended and combined into a single tube per sample. Centrifugation was repeated for 10 min, followed by an additional wash with 10 mL PBS, and finally with 5 mL deionized water. The pellets were resuspended in 2 mL deionized water. The washed samples were transferred to pre-weighed quartz pressure vials and centrifuged 5 min. The supernatant was carefully removed with a micropipettor as much as possible without disturbing the pellet. The sample vials were placed in a vacuum oven at 85 °C for 48 h [44]. After cooling the samples to room temperature, the samples were weighed 6 times, to determine the average dry weight. Samples were submitted to Dr. Jong Sung Kim, Health and Environments Research Centre (HERC), Dalhousie University, for trace element quantification. For this, a microwave-assisted digestion was used to homogenize bacterial samples using a Discover SP-D microwave digester (CEM Corporation, Matthews, NC, USA). To samples in 10 mL quartz pressure vials, 600 µL of concentrated nitric acid (trace metal grade nitric acid, Thermo Fisher Scientific, Waltham, MA, USA) and 450 µL of hydrogen peroxide were added. Digested samples were diluted with Milli-Q water to obtain approximately 2% nitric acid concentration before analysis. An iCAP Q inductively coupled plasma mass spectrometer (ICP-MS, Thermo Fisher Scientific, Waltham, MA, USA) paired with an ESI SC-4DXS autosampler (Elemental Scientific, Omaha, NE, USA) was used to measure iron, manganese, and zinc. All samples were run in kinetic energy discrimination mode, using high purity helium (99.999%) as the collision gas.

## 5. Conclusions

In the present study, we determined that the iron chelator DIBI is able to improve the outcome of ciprofloxacin treatment of *S. aureus* strains by suppressing growth during the recovery phase of an in vitro killing assay. It was found that most bacterial colonies plated following 72–96 h CIP exposure were in the form of a small colony variant morphology, which was not observed when DIBI was combined with CIP. When these SCV clones were re-grown in the absence of antibiotic, the slow growth phenotype remained, and these stable mutants were found to have an elevated MIC to fluoroquinolones. In contrast, clones of *S. aureus* that were exposed to CIP in the presence of DIBI did not have altered MICs. The data presented herein support the hypothesis that iron chelation by DIBI in combination treatment can suppress mechanisms that produce spontaneous resistance to CIP, adding to the body of evidence that DIBI could be useful as an adjunct antimicrobial. Further molecular genetic testing is required to confirm and describe the alterations in the CIP-exposed variants, and to determine the precise mechanism that produced them. As this study is limited to in vitro experiments, this phenomenon should be explored in an appropriate in vivo model for further significance.

## Figures and Tables

**Figure 1 antibiotics-11-01642-f001:**
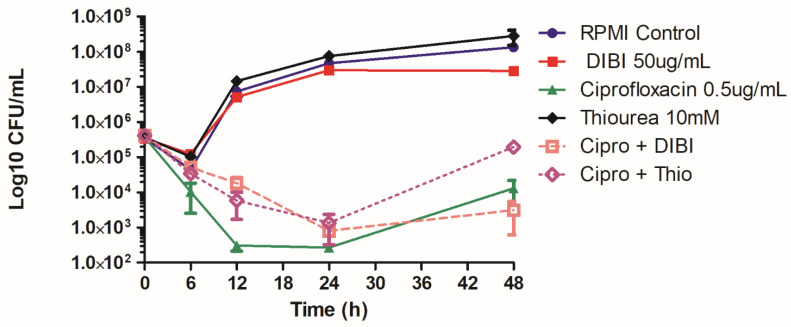
Effects of ciprofloxacin, DIBI and thiourea on *Staphylococcus aureus* killing and recovery growth. ATCC 25923 was grown in MHB and a standardized inoculum was added to 10 mL flask cultures of untreated RPMI control (●), RPMI with added 50 µg/mL DIBI (▪), 0.5 µg/mL CIP (▲), 10 mM thiourea (♦), 50 µg/mL DIBI combined with 0.5 µg/mL CIP (□) and 10 mM thiourea combined with 0.5 µg/mL CIP (◊). Samples obtained during exposure were plated onto BA and CFU/mL was determined. The data represent means ± SEM for three independent experiments.

**Figure 2 antibiotics-11-01642-f002:**
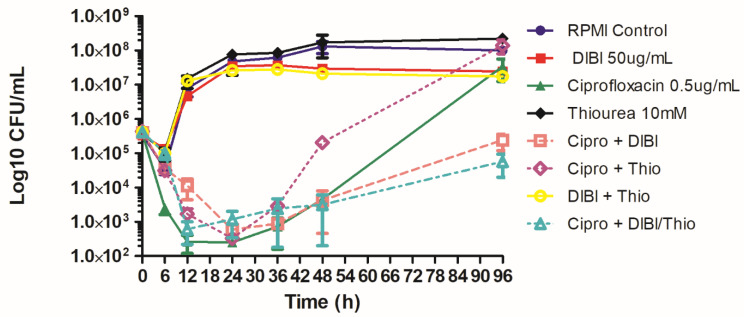
Long term effects of ciprofloxacin, DIBI and thiourea on *Staphylococcus aureus* killing and recovery growth. ATCC 25923 was grown in MHB and a standardized inoculum was added to 10 mL flask cultures of untreated RPMI control (●), RPMI with added 50 µg/mL DIBI (▪), 0.5 µg/mL CIP (▲), 10 mM thiourea (♦), 50 µg/mL DIBI combined with 0.5 µg/mL CIP (□), 10 mM thiourea combined with 0.5 µg/mL CIP (◊), 50 µg/mL DIBI combined with 10 mM thiourea (ο) and 10 mM thiourea combined with 50 µg/mL DIBI and 0.5 µg/mL CIP (Δ). Samples obtained during exposure were plated onto BA and CFU/mL were determined. The data represent means ± SEM for two independent experiments.

**Figure 3 antibiotics-11-01642-f003:**
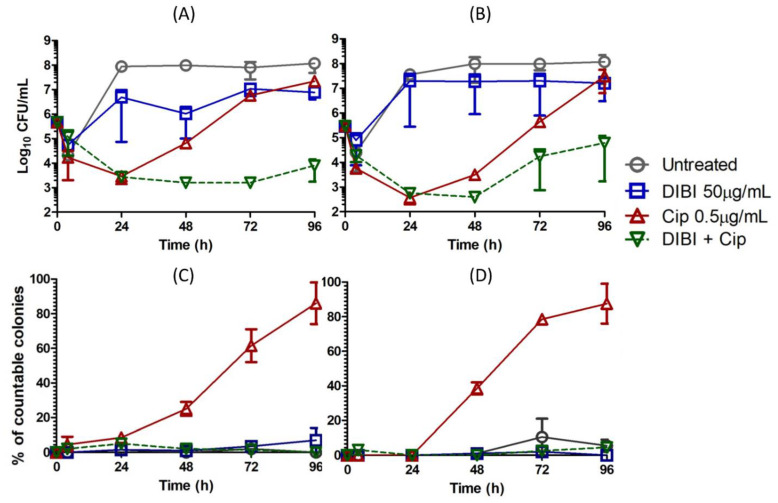
Effects of ciprofloxacin and DIBI on *Staphylococcus aureus* killing and small colony variant recovery growth. *S. aureus* ATCC 43300 (**A**,**C**) and ATCC 25923 (**B**,**D**) were grown in MHB and a standardized inoculum added to 10 mL flask cultures of untreated RPMI control (ο), or RPMI with added 50 µg/mL DIBI (□), 0.5 µg/mL CIP (Δ) or 50 µg/mL DIBI combined with 0.5 µg/mL CIP (▽). Samples taken during exposure were plated onto BA for enumeration of total CFU/mL (**A**,**B**) and the percentage of SCVs (**C**,**D**) in the recovered populations. The data represent means ± SEM for two independent experiments.

**Figure 4 antibiotics-11-01642-f004:**
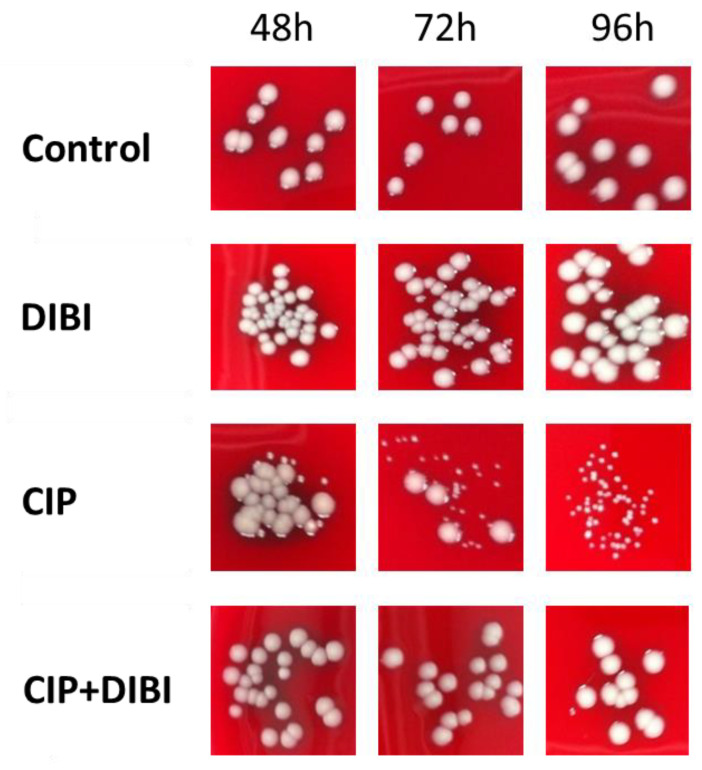
Effects of ciprofloxacin and DIBI on *Staphylococcus aureus* small colony variant formation. *S. aureus* ATCC 25923 was grown in MHB and a standardized inoculum added to 10 mL flask cultures of untreated RPMI control, RPMI with added 50 µg/mL, 0.5 µg/mL CIP or 50 µg/mL DIBI combined with 0.5 µg/mL CIP. Samples taken during exposure were plated onto BA for enumeration of total CFU/mL and the percentage of SCV in the recovered population. All pictures are at the same final magnification and representative of two independent experiments.

**Table 1 antibiotics-11-01642-t001:** Influence of growth medium on bacterial cellular metal content.

Growth Medium	Medium Metal µM			*S. aureus* Cellular Metal µM								
				ATCC 43300			ATCC 25923			ATCC 6538		
	Fe	Mn	Zn	Fe	Mn	Zn	Fe	Mn	Zn	Fe	Mn	Zn
MHB	6.420	0.300	11.067	3.700	0.460	0.828	4.730	0.440	0.953	7.310	0.690	1.279
FECMHB	0.250	0.004	2.634	0.810	0.020	1.012	1.014	0.020	0.909	1.140	0.030	1.087
RPMI	0.080	0.025	0.559	0.390	0.040	0.959	0.610	0.030	0.605	0.570	0.040	1.057

MHB; Mueller-Hinton Broth, FECMHB; MHB deferrated with iron chelator FEC-1, RPMI; Roswell Park Medical Institute cell culture medium.

**Table 2 antibiotics-11-01642-t002:** Effect of growth medium on *Staphylococcus aureus* sensitivity to ciprofloxacin and DIBI.

Cultivation Medium ^a^	ATCC 25923		ATCC 43300	
	CIP MIC µg/mL (µM)	DIBI MIC µg/mL (µM)	CIP MIC µg/mL (µM)	DIBI MIC µg/mL (µM)
MHB	0.5 (1.5)	2000 (222)	0.5 (1.5)	8000 (888)
FECMHB	ND	1 (0.11)	ND	2 (0.22)
RPMI	0.25 (0.75)	2 (0.22)	0.25 (0.75)	2 (0.22)

^a^ Initial inoculum cultivation medium; washed inoculum tested in RPMI for all, MIC values are from 24 h reads; MHB; Mueller-Hinton Broth, FECMHB; MHB deferrated with iron chelator FEC-1, RPMI; Roswell Park Medical Institute cell culture medium MIC; minimum inhibitory concentration, CIP; ciprofloxacin.

**Table 3 antibiotics-11-01642-t003:** Colony counts of ciprofloxacin-treated *Staphylococcus aureus* ATCC 25923 in replete and deferrated MHB.

Culture Medium	CFU/mL	
	0 h	24 h
MHB	1 × 10^5^	3 × 10^7^
MHB+2 × MIC CIP	1 × 10^5^	2 × 10^1^
FECMHB	7 × 10^4^	3 × 10^7^
FECMHB +2 × MIC CIP	7 × 10^4^	6 × 10^1^

MHB; Mueller-Hinton Broth, FECMHB; MHB deferrated with iron chelator FEC-1, MIC; minimum inhibitory concentration, CIP; ciprofloxacin.

**Table 4 antibiotics-11-01642-t004:** Median 24 h MIC values for 96 h ciprofloxacin kill/growth *Staphylococcus aureus* survivor clones.

Treatment	DIBI ^a^ µg/mL (µM)	CIP ^a^ µg/mL (µM)	MOX ^b^ µg/mL (µM)	MUP ^b^ µg/mL (µM)	GEN ^c^ µg/mL (µM)	VAN ^c^ µg/mL (µM)
Control	4 (0.4)	0.125 (0.38)	0.062 (0.15)	0.062 (0.123)	0.062 (0.13)	0.5 (0.345)
DIBI	4 (0.4)	0.125 (0.38)	0.062 (0.15)	0.062 (0.123)	0.062 (0.13)	0.5 (0.345)
CIP	4 (0.4)	1.00 (3.0)	0.25 (0.62)	0.062 (0.123)	0.062 (0.13)	0.5 (0.345)
CIP + DIBI	4 (0.4)	0.125 (0.38)	0.062 (0.15)	0.062 (0.123)	0.031 (0.065)	0.5 (0.345)

MIC; minimum inhibitory concentration, DIBI; iron-chelating polymer, CIP; ciprofloxacin, MOX; moxifloxacin, GEN; gentamicin, MUP; mupirocin, VAN; vancomycin; ^a^ ATCC43300 and ATCC25923 tested; ^b^ only ATCC 43300 tested; ^c^ only ATCC 25923 tested.

## Data Availability

The data presented in this study are available on request from the corresponding author.

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
