# Peer review of "Iron Chelator DIBI Suppresses Formation of Ciprofloxacin-Induced Antibiotic Resistance in Staphylococcus aureus"

_antibiotics, 2022, doi:10.3390/antibiotics11111642_

Round 1

Reviewer 1 Report

The manuscript presented for review is well-written and structured report of the iron chelator DIBI suppresses formation of ciprofloxacin induced antibiotic resistance in Staphylococcus aureus. Multidrug-resistance Staphylococcus aureus strains pose a serious treatment problem among hospitalized patients. Moreover, CA-MRSA infections are now becoming more common. Therefore, the subject of presented manuscript raises an especially important problem.

The manuscript is interesting, complete and well structured, but it is necessary to do some major revision to be accepted:

Major specific suggestion/comments:

Please make sure the affiliation is correct.

In the text, reference numbers should be placed in square brackets [ ], and placed before the punctuation; for example [1], [1–3] or [1,3]. Please correct throughout the text.

I suggests shortening the introduction.

Please add conclusion section

Line 16: was similar to à was like

Line 49: a number of à several

Line 61: Staphylococcus aureus (S. aureus) à please explain all abbreviations before using them

Line 76: all of à all

Line 89: Please use full names in titles

Line 94: has ability to à can

Line 115: please enter a space

Line 117: 2000-4000 fold à 2000-4000-fold

Line 130: please enter a space

Line 135: i.e. à i.e.,

Line 140: i.e. à i.e.,

Line 168:  experiments, but à experiments but

Line 170: in spite of à despite

Line 176: similar to à like

Line 256: tested, and à tested and

Line 266: topoisomerase à topoisomerase,

Line 282: similar to à like

Line 297: i.e. à i.e.,

Line 365: 4.1 à please add numbering throughout the section

Line 412: please remove the double space

Author Response

Thank you for your constructive comments which have been used to provide a revised m/s

Reviewer 2 Report

In this manuscript of "Iron chelator DIBI suppresses formation of ciprofloxacin-induced antibiotic resistance in Staphylococcus aureus", the authors found the iron chelator DIBI can not promote resistance to itself for Staphylococcus aureus. However, DIBI totally suppressed SCV formation with all survivors remaining sensitive to CIP and to DIBI. All the results indicated that DIBI is  potential as an adjunct to CIP and other antibiotics to improve antibiotic efficacy and to thwart antibiotic resistance development.

The manuscript is well written. However, some issues should be addressed for further consideration.

1. Through the manuscript, the manganese and zinc content in Table 1 made non-sense. Is that possible that DIBI could chelate with these two metals? It will make the reader misunderstand.

2. What is the purpose for S. aureus ATCC 6538 in Table 1? There is no research and discussion about this bacteria. Maybe it could be deleted.

3. In Section 4, Materials and Methods, the last part should be "trace element quantification for iron, manganese, and zinc" for Table 1, but not Iron Measurements only.

4. The body paragraph from line 210 to 215 should be the notes of Figure 3, as well as line 224 to 229 for Figure 4.

5. Please present Figure 3 with higher resolution.

Author Response

Thank you for your constructive comments which we have utilized to revise our m/s 

Reviewer 3 Report

The manuscript entitled: Iron chelator DIBI suppresses formation of ciprofloxacin-in- 2induced antibiotic resistance in Staphylococcus aureus by the authors David S. Allan and Bruce E.  Holbein have done a great job by determining the ciprofloxacin-in- 2 induced antibiotic resistance in Staphylococcus aureus due to Iron chelator DIBI suppresses formation .There are some flaws that shall be addressed. some of them are:

1.DIBI shall not be used in Tittle of manuscript.

2.Line -2 of abstract severely limits means??

3. Add   conclusion and recommendations of the manuscript at the end.

4. Also add limitations of yours study.

5. Yet I am not a native speaker of English language but still I recommend that the English language needs touching up in a major way. The article needs to be rewritten in readable English. Many sentences are confusing, do not lead to scientific meaning, and can be found starting in lower case, and upper case can be detected in the middle of sentences without proper nouns.

Author Response

Thank you for your constructive comments which we have used to revise our m/s 

Round 2

Reviewer 1 Report

Dear Authors,

The revision of the manuscript has answered all of the questions .

Reviewer 2 Report

The authors have done all the necessary correction and now the manuscript could be accepted in this current version.

Reviewer 3 Report

The authors have incorporated the suggested changes. I recommend the article for publication in antibiotics.